# Turning Tool Wear Estimation Based on the Calculated Parameter Values of the Thermodynamic Subsystem of the Cutting System

**DOI:** 10.3390/ma14216492

**Published:** 2021-10-29

**Authors:** Victor Petrovich Lapshin

**Affiliations:** Department of Automation of Production Processes, Faculty of Automation, Mechatronics and Control, Don State Technical University, 344002 Rostov-on-Don, Russia; lapshin1917@yandex.ru

**Keywords:** cutting system, thermodynamics, tool wear, vibrations

## Abstract

Today, modern metalworking centers are not yet able to reliably assess the degree of wear of the tool used in metal cutting. Despite the fact that a large number of methods for monitoring the service life of the tool have been developed, this issue still remains a difficult task that needs to be solved. Idea: The article proposes a new, previously unused method for estimating the power of a cutting wedge in metalworking. The aim of the study is to develop a method for indirectly estimating the tool wear rate based on a consistent model of intersystem communication that describes the force, thermal and vibration reactions of the cutting process to the shaping movements of the tool. Research methods: The study consists of experiments on a measuring stand and a homemade measuring complex. It also uses the Matlab mathematical software package for processing and graphical interpretation of data obtained during experiments. The results show that the proposed method of estimating the current tool wear is applicable for the interpretation of experimental data. Statistically, the modified Voltaire operator of the second kind models the temperature more accurately; at the peak, this method is three times more accurate than the other.

## 1. Introduction

Global manufacturers of metal-cutting equipment, such as, Siemens and Fanuc Co., Ltd. (Munich, Germany), use the cutting-edge technologies developed in the world today. A metalworking center produced by a brand such as Fanuc Co., comparing in the degree of automation and intellectualization of the control system, is highly competitive with systems such as a control system for a modern aircraft or even a spacecraft. Now, the development of electronics and programming makes it possible to solve the problems of cutting control on metal-cutting machines at a new, very high level. Talking about this new level, we principally mean a widespread application of digital control algorithms based on a complex mathematical apparatus describing the interrelated dynamics and evolution of the cutting process. In case of the dynamics of cutting processes, this is a well-developed topic in which the assessment of the vibration activity of the tool under cutting plays a big role [1,2,3,4,5]. In [1,4], the issue of assessing the degree of wear of the cutting tool by the measured force response from the cutting process to the shaping movements of the tool is considered. In [2,4], the authors consider the issues of vibration diagnostics of critical wear of the cutting tool, according to the calculated values obtained from the signal removed by the vibration accelerometer. It is also possible to assess tool wear by a vibroacoustic signal taken through a microphone [3]; however, such a procedure is not possible in cases of cutting using a cooling system of the cutting zone. Work [5] is devoted to the evaluation of the relationship between the vibration dynamics of the cutting system and the tool wear process. Here, the authors confirm the well-known fact that with the increase in the wear of the instrument, the amplitude of its vibrations increases; however, in their last 40 passes they could not confirm this statement, which indicates a low degree of reliability of the procedure for vibration diagnostics of wear processes. Proceeding from this, the assessment of wear by the measured dynamics of the cutting process (tool vibration), as well as from the analysis of the rigidity of the cutting system, reflected in the measured force reaction to the shaping movements of the tool, cannot guarantee high reliability of the estimated wear parameter.

As part of the widespread adoption of Industry 4.0 tools, including in metalworking on cutting machines, assessment of vibration activity under cutting becomes the cornerstone around which modern systems for diagnosing the cutting process are built. However, in addition to providing low vibration activity when cutting metals, its connection with the quality of the products obtained is taken into account. The second major challenge in the quality of treatment processes is the control and support of the specified processing temperatures [6]. Considering that the vibration activity of the tool and the cutting temperature are related to each other through the evolution (wear) of the cutting wedge under metal processing, a third task arises. It is assessing the residual tool life and/or providing the maximum value of this parameter [7,8,9,10]. In modern machine-building production, a tool is just a consumable material that must be replaced when its cutting properties are exhausted. In this regard, a more important task is to predict the residual tool life when performing complex and bulk operations [11]. The relationship between tool wear and changes in its dynamic properties is well described in the works of V. P. Astakhov [12], where the correlation of vibrations and temperature with the evolution of the cutting wedge is shown. However, predicting the development of the wedge wear under metal-cutting is quite a difficult task, and thus, intelligent systems and models, which provide assessing the current wear rate by indirect measurable signs, are widely used [13,14,15,16].

Here, by intelligent systems, we mean systems that have humanlike data processing algorithms, such as fuzzy algorithms and/or algorithms built on a neural network approach. Despite the fairly good results of using such systems for forecasting and assessing the tool wear rate in metalworking, there are significant limitations with the widespread implementation of this approach in mass production. This limitation is due to the fact that in each new machine, the complex cutting dynamics will be unique, and the application of a machine-specific intuitive diagnostic algorithm becomes impossible on another machine.

The solution to this problem, in our opinion, is in the application of the existing scientific approaches to the description of the interconnected complex, nonlinear dynamics of the cutting process, taking into account the regenerative nature of self-excitation of vibrations under processing [17,18,19,20,21]. At the same time, we understand that this dynamic in the cutting process state space has pronounced nonlinear properties [22], including points of the bifurcation (rebuilding) of the cutting system [23].

As for the disadvantages of previous studies carried out in this area, it should be noted that there are two main directions of development of scientific thought in the field of mathematical modeling of temperature effects when cutting metals in metal-cutting machines. The first such direction is the direction in which the temperature in the contact zone of the tool and the workpiece has a static design value, for example, as in [6,7], determined by static design values of forces and processing speeds. As a result of this approach, there is a gap between the subsystems of the vibrational activity of the tool [1,2,3,4,5] and the thermodynamics of the cutting process [6,7]; however, experimental studies show that such a relationship exists [9,10]. The second direction is related to the modeling of the temperature field in the tool itself, this approach is very well presented in [8], and it should be noted that this approach quite accurately connects the development of wear of the cutting wedge with the temperature gradient in the same cutting wedge. However, this approach has low practical application, since it is practically impossible to measure the gradient of the temperature field in the cutting wedge during cutting, and the interrelated dynamics of the cutting process and its relationship with the temperature are also not taken into account here. Therefore, within the framework of these models, the solution to the problem associated with the wear resistance of the tool is the preliminary preparation of the cutting tool; that is, giving the tool material, due to, for example, spraying, such temperamental properties at which its wear rate will be small. The disadvantage of both of these approaches is that it is not possible to use modern measuring systems for cutting monitoring to increase the efficiency of the entire cutting system, and this is despite the fact that these measuring systems are developing at a gigantic pace today.

In this regard, as the purpose of the study, the development of a new method for assessing the degree of wear of a cutting tool was chosen, which would rely on a mathematical apparatus describing the interrelated measurable dynamics of the cutting process on metal-cutting machines.

As an assessment of the novelty of the research and the scientific contribution, I note that the approach proposed in this work has not been used by anyone before; that is, the research is innovative in nature. In general, the impetus for the formation of a named modeling method and an approach based on it to assess the degree of tool wear by the parameters of the thermodynamic subsystem is the rapid development and implementation of measuring systems for monitoring the dynamics of processes in metal-cutting machines. That is, in other words, previously such a mathematical model was not needed, so it was impossible to measure and process data on the dynamics of temperature and vibrations of the processing process in real time. These arguments allow us to speak about the high scientific and practical significance of the research.

## 2. Materials and Methods

To synthesize a mathematical model of the relationship between wear, the measured temperature in the tool–workpiece contact zone, the cutting force and the vibration activity of the tool—we conducted a series of full-scale experiments on the 1K625 lathe (see Figure 1a) with the STD. 201-1 stand installed on it for investigating cutting modes during turning (see Figure 1b), as well as experiments on a self-engineered experimental complex (see Figure 1c,d). The measuring stand STD. 201-1 allows you to measure the force reaction spread out along the axes of the tool deformation, the temperature in the cutting zone by using the effect of natural thermal EMF formed in the contact of the tool and the workpiece. The measuring stand also includes three vibration accelerometers that measure vibrations along the axes of deformation of the instrument. The experimental machine was previously upgraded, the motor control, which provides operating modes on the machine, was switched to frequency control. Thanks to this, it became possible to smoothly adjust the cutting speed inside the selected operating mode of the machine.

The 1K625 machine, shown in Figure 1, was upgraded; specifically, the motor control, providing operation modes on the machine, was transferred to frequency regulation. The frequency convertor is presented in the upper right corner of Figure 1a. As a result, it became possible to smoothly adjust the cutting speed within the selected machine operation mode. The mode parameters in the experiments were as follows: cutting speed *V_c_* = 124 m/min, feed *fr* = 0.11 mm/rev, cutting depth *a_p_ =* 0.1 mm for the entire period of the experiment, the cutting parameters remained unchanged. The choice of these parameters is related to the capabilities of the machine itself, for which, taking into account the processed material, these are the most optimal cutting elements. As a tool, we used MR TNR 2020 K11 holder and a pentahedral plate 10113-110408 T15K6 as a cutter on it, as well as WNUM 120,612 (02114-120612) H30 (T5K10) KZTS—a replaceable hex plate “broken triangle”, cutting plate with an angle at the top (angle of attack) *γ*_0_ = 35°, and the main angle in the plan *φ* = 80° (the angle between the projection of the main cutting edge on the main plane and the feed direction), both holders and cutting plates are made in Russia (Yelets city, Barkovsky str., 3, of 2). In both experiments, the holder was isolated from the machine support and the inner walls of the stand, using textolite plates. The shaft made from steel grade 45 (gost) was machined, this steel grade is widely used both in Russia (GOST 2591-2006–Steel 45 (st45)—structural carbon is of high quality), and in the USA and Germany, the American standard ASTM A568M marks this type of steel as AISI 1045, and in Germany the DIN17200 standard is Ck45. Steel 45 (st45) contains from 0.42 to 0.5% carbon (this can be traced in the name of the steel grade), 97% iron, as well as percentages of silicon, manganese, nickel, sulfur, phosphorus, chromium, copper, arsenic [24,25,26]. The shaft, made of 45% steel, was made using hot rolling technology; before the experiment, the shaft was cut to a length of 75 cm, then there was a pre-finishing treatment of the shaft surface and precise alignment. The choice of this steel grade is associated with its widespread use in mechanical engineering, which is due to the high quality of structural steel and its relatively low price in the steel market. In total, more than 15 series of experiments were carried out; the article presents only part of the results of these experiments. In the second experiment, vibration transducers manufactured by GLOBAL TEST AR2081-10 (Sarov, Russia) and connected to the GLOBALTEST AR13 cable were used. These vibration converters have an analog output with a signal, having a very high natural cutoff frequency of 48 kHz; the processing process itself has a basic vibration frequency in the range from 1 to 4 kHz. To digitize such a signal, it is necessary to have a quantization frequency of at least 8 kHz; in this regard, an ADC of the company L-CARD (Moscow, Russia) E14-440 AD/DA CONVERTER with a USB 2.0 cable (USB Type B) was used. The experiment itself and its results are also described in our previous articles, which consider the interpretation of the results by other methods [23,27].

The experimental complex developed by us (see Figure 1c,d) contains three vibration accelerometers and an artificial thermocouple built into the cutting wedge close to the contact zone of the back edge of the tool and the workpiece. To embed the thermocouple in the cutting wedge, it was pre-cut through the electroerosion metal cutting, and the thermocouple was hot melted in the prepared hole.

Tool wear was estimated in the experiment by the back edge. For this, after each processing step, a photo of the cutting wedge was taken; examples of these photos are shown in Figure 2.

Figure 2a,b is enlarged photographs of the cutting plate that has undergone preliminary running-in, and the cutting plate with critical flank wear. It can be seen that the flank wear rate has changed from almost zero to a value close to 0.45 mm. From Figure 2c,d, it can be seen that the experiment was also carried out from the moment of running-in until the formation of a significant tool flank wear area.

## 3. Basic Mathematical Model Synthesis

As for modeling the temperature in the contact zone of the tool with the workpiece, I proceeded from the hypothesis of temperature inheritance of the results of temperature isolation along the entire cutting path at the point of contact. That is, the temperature value at a given time and in a given tool–workpiece contact zone will be determined not only by the current value of the heat generated here but also by the influence of the power of irreversible transformations along the entire previous path of the product processing time. Assume that *N(n)* is the current value of the power of irreversible transformations, *L(n)* is the current value of the path traveled by the tool during machining, and *t(n)* is the current value of the processing time. Then, at each point of the workpiece processed surface passed by the tool, a certain amount of heat will be released proportional to the power of irreversible transformations. The current temperature value in the zone under consideration will be determined both by the value of heat quantity released here, and by the influence of the previously released heat along the path traveled by tool (*L*), taking into account the time of energy dissipation (*t*). As a basic model for describing this phenomenon, I considered the modernized Volterra operator, represented as the following double integral [27]:(1)$$Q_{z}=Q_{s}+k_{T}\iint_{D}w_{L}(\xi -L)w_{t}(\eta -t)N(\gamma ,\eta )d\gamma d\eta$$
where $Q_{z}$—the temperature value at the point of the tool—workpiece contact, $Q_{s}$—the ambient temperature, $w_{L}(\xi -L)$—the kernel of the Volterra operator, which characterizes the dissipation of power converted to temperature in the space given by the geometric shape of the part, $w_{t}(\eta -t)$—the kernel of the Volterra operator, which characterizes the dissipation of power converted to temperature in the time set by the processing program. To be more specific, let’s consider the mechanism of temperature transmission through the back edge of the tool when turning metals on metal-cutting machines (see Figure 3) [9].

As can be seen from Figure 3, the back edge, formed under machining, contacts during cutting with that part of the workpiece that will be turned through the spindle rotation period. As a rule, the temperature in the tool–workpiece contact zone stabilizes within seconds after the start of cutting. During this time, the tool warms up; a temperature field whose gradient is directed towards the tool is formed; the temperature flow is directed to the part of the workpiece that will be turned through the spindle rotation period, that is, in the direction of the formed tool–workpiece contact area (in Figure 3a, it is the zone indicated by h_3_, where h_3_ is the depth of this area in mm). Through this, the cutting zone is preheated due to the temperature released earlier under cutting. Thus, all temperature previously allocated during machining affects the current temperature in the tool–workpiece contact zone through the connection formed by the contact area along the back edge. It is this process that can be described using a mathematical apparatus based on the Volterra operator of the second kind. Therefore, it will take, as a basic model, a multiplicative criterion for assessing the influence of the previous increase in temperature on its current value in the form of the following double integral [27]:(2)$$Q_{z}=Q_{s}+k_{Q}\int_{0}^{L(t)}e^{\frac{\alpha_{1}}{\lambda }(\xi -L)}d\xi \int_{0}^{t}e^{\frac{\alpha_{2}}{T_{h}}(\eta -t)}N(\eta )d\eta$$
where *α*_1_, *α*_2_—dimensionless scaling parameters of the integral operator to be identified, λ—the coefficient of thermal conductivity, $Q_{z}$—the temperature value in the tool–workpiece contact zone, $Q_{s}$—the ambient temperature, $k_{Q}$—the coefficient characterizing the conversion of the power of irreversible transformations allocated in the tool—workpiece contact zone into temperature, $L(t)=V_{c}t$—the path traveled by the tool during cutting, *V_c_*—the cutting speed in mm/s, N(η)—the power allocated in the tool–workpiece contact under cutting. To describe the power released in the cutting zone, consider the diagram of the decomposition of the force response from the cutting process to the movements of the shaping tool along the axes of deformation of this tool during turning (see Figure 4).

In the diagram (Figure 4), the decomposition of deformations into three main axes is accepted: *x*-axis—the axial direction of deformations (mm), *y*-axis—the radial direction of deformations (mm), and *z*-axis—the tangential direction of deformations (mm). Along the same axes, the force response is decomposed from the cutting process to the shaping motions of the tool (*F_f_, F_p_, F_c_ (N)*), *V_f_* and *V_c_* (mm/s) of the feed and cutting speeds, respectively, ω—the angular spindle speed (rad/s).

The relationship between force components $F_{f},F_{p},F_{c}$ depends on many factors, such as, the geometry of the cutter, the cutter wear rate, etc. [28]. So, in [29], when machining with a sharp cutter with the main tool rake angles *γ*_0_ = 35°, *φ* = 80°, the ratio between the components is on average equal to: (3)$$F_{f},F_{p},F_{c}=(0.3-0.4),(0.4-0.5),(1)$$

Taking into account the diagram shown in Figure 4, we represent the power of reversible transformations as:(4)$$N=\sqrt{{\left(F_{c}\right)}^{2}+{\left(F_{p}\right)}^{2}+{\left(F_{f}\right)}^{2}}\sqrt{{\left(V_{f}-\frac{dx}{dt}\right)}^{2}+{\frac{dy}{dt}}^{2}+{\left(V_{c}-\frac{dz}{dt}\right)}^{2}}$$
where $F_{f},F_{p},F_{c}$—the components of the force response formed on the front edge of the tool, $V_{f},V_{c}$—speeds set by the CNC program, the feed rate and the cutting speed, respectively, $\frac{dx}{dt},\frac{dy}{dt}.\frac{dz}{dt}$—speeds of the deformation motions of the tool.

Based on the analysis, we formulate the idea of a mechanism for the mutual influence of force and temperature in the cutting zone, wear and vibrations of the cutting tool, which is convenient to perform by building feedbacks in the cutting process. Thus, we obtain a system consisting of the following subsystems:-a mechanical subsystem, or a subsystem that forms a force response to the shaping motions of the tool;-a thermodynamic subsystem responsible for the formation of temperature in the tool–workpiece contact zone;-a deformation subsystem that describes the dynamics of deformation motions of the tool;-tool evolution (wear), a subsystem describing the process of tool wear under cutting.

Taking into account the previously identified features of the processes, I will construct the following structural and logical scheme of the relationship between the introduced subsystems (see Figure 5). Here, I note that the relationships presented in this diagram are not permanent; that is, they are not rigidly defined but change by strengthening or weakening as the processing subsystems that form them develop or degrade.

Summarizing the scheme of the processing process structure shown in Figure 5, I will describe the evolution of the tool during cutting as follows: tool formation–workpiece contact area serves the purpose of self-organization of the cutting system through the formation of additional thermodynamic feedback, whose stabilization is provided by a certain combination of tool wear and a limited vibration cutting mode. In other words, during machining, the cutting system seeks to run in the cutting wedge of the tool in such a way as to reach a certain wear level that reduces the vibration activity of the tool and provides the thermodynamic feedback. This, in turn, enables, through the stabilization of the force response, to achieve the maximum possible reduction in the further intensity of the cutting wedge wear. Based on these considerations, as well as the analysis of the interconnectedness of the subsystems presented in Figure 5, it can be concluded that all subsystems are closed to the thermodynamic one, which can serve as an indicator of the machining process and enables to indirectly judge the degree of tool wear. It should be noted that in the cutting zone, the influence of temperature on the power subsystem and, as a consequence, on the vibration subsystem of the cutting system is a well-studied factor today [30].

Turning back to Equation (2), summarizing the above arguments about the thermodynamic subsystem of the cutting system, reveal such an indicator of this equation as $T_{h}$. This indicator is some constant that has the dimension of time and determines the interaction time of the tool back edge and the workpiece. This constant, based on the processed data and the observation made earlier, is directly proportional to the flank wear rate and inversely proportional to the vibration energy of the tool in the cutting direction. That is, it has the following form: (5)$$T_{h}=\frac{h_{3}}{VA}$$
where *VA*—the energy of the vibration signal calculated from the following formula:(6)$$VA=\sqrt{\frac{1}{T_{v}}(\int_{0}^{T_{v}}{\frac{dy}{dt}}^{2}dt})$$
here, VA—can be interpreted as the signal background noise, or the energy of the vibration signal over the observation period (experiment)—$T_{v}$, *h_3_*—tool flank wear. In fact, the greater the wear and the lower the vibration activity of the tool, the greater this constant is. In general, the effect of the introduced constant on the temperature in the processing zone can be described as follows: the higher the value *T_h_*, the stronger the effect on the current temperature in the tool–workpiece contact zone, previously selected under cutting temperature; the lower *T_h_*, the less such influence is. In other words, the greater *T_h_*, the higher the temperature in the processing zone. Similar arguments are valid for λ—the thermal conductivity coefficient. No additional explanation is required here since this coefficient is directly included in the Fourier equation, and the greater this coefficient, the stronger the influence of the temperature gradient on the amount of heat flow directed into the workpiece.

Integral operator (2) has a solution for the case when the power of irreversible transformations is a constant value $N_{0}=N(t)$.
(7)$$Q_{z}=Q_{s}+\frac{k_{Q}N_{0}\lambda T_{h}}{\alpha_{1}\alpha_{2}V_{c}}(1-e^{-\frac{\alpha_{{}_{1}}V}{\lambda }t})(1-e^{-\frac{\alpha_{2}}{T_{h}}t})=Q_{s}+\frac{k_{Q}N_{0}\lambda h_{3}}{VA\alpha_{1}\alpha_{2}V_{c}}(1-e^{-\frac{\alpha_{{}_{1}}V_{c}}{\lambda }t})(1-e^{-\frac{\alpha_{2}VA}{h_{3}}t})$$

Equation (7) is essentially a solution to a standard second order linear differential equation, such as the one below:(8)$$Q_{z}=Q_{s}+kN(1-e^{-\frac{t}{T_{1}}})(1-e^{-\frac{t}{T_{2}}})$$
or the differential equation itself:(9)$$T_{1}T_{2}\frac{d^{2}Q_{z}}{dt^{2}}+(T_{1}+T_{2})\frac{dQ_{z}}{dt}+Q_{z}=kN$$
where $T_{1}=\frac{\lambda }{\alpha_{1}V_{c}}$, $T_{2}=\frac{T_{h}}{\alpha_{2}}=\frac{h_{3}}{VA\alpha_{2}}$, $k=\frac{k_{Q}\lambda h_{3}}{VA\alpha_{1}\alpha_{2}V_{c}}$—transfer factor.

The Equation obtained in Equation (9) is a mathematical model of the thermodynamic subsystem of the cutting system with some previously offered assumptions. Here, in Equation (9):(10)$$T=\sqrt{T_{1}T_{2}}=\sqrt{\frac{\lambda h_{3}}{VA\alpha_{1}\alpha_{2}V}}$$

The time constant of this thermodynamic subsystem of the cutting system. As can be seen from Equation (10), this constant includes data obtained from the subsystem of vibration motions of the tool (VA), and data from the subsystem reflecting the wear rate (*h*_3_). In terms of the modeling equation (9), constant (10) reflects the slope of the transition characteristic of the thermodynamic subsystem of the cutting system. In other words, the time constant of a thermodynamic subsystem determines the response rate on the part of this subsystem to a change in the power of irreversible transformations. The temperature in the tool–workpiece contact, when turning metals on metal-cutting machines, is measured as follows: using an artificial thermocouple built into the tool (see Figure 1c), and through natural thermoelectric emf effect, taken by the measuring stand STD.201-1 (see Figure 1b). In this case, it is possible to obtain a curve for increasing the cutting temperature, the rate of change of which will be correctly estimated, even if these two measurement methods are imperfect in accuracy.

## 4. Discussion and Results

The results of the experiments carried out using the measuring stand STD.201-1 are shown in Figure 6.

The test results are shown in the graphs of the cutting force Figure 6a–c, decomposed along the axes of deformation, and the temperature rise graph in the cutting zone Figure 6d. In addition to these data, I obtained the results of measuring the vibration activity of the tool along the axes of deformation. On account of these values, I calculated discrete values of the power of irreversible transformations using Equation (4). The data obtained in the experiment make it possible to obtain a graph of calculated temperature values using the integral operator (2). However, this operator cannot be used directly since these data are discrete (digital) in nature. Based on the obtained mathematical apparatus (see Equation (2)), I will represent the integral operator in discrete form, as it is represented in the Equation (10):(11)$$Q_{z}{}^{n}=\frac{k_{Q}e^{-\frac{\alpha_{1}}{\lambda }L(t_{n})}e^{-\frac{\alpha_{2}}{T_{h}}t_{n}}\lambda T_{h}}{\alpha_{1}\alpha_{2}}[N_{2}(e^{\frac{\alpha_{1}}{\lambda }L(t_{2})}-e^{\frac{\alpha_{1}}{\lambda }L(t_{1})})(e^{\frac{\alpha_{2}}{T_{h}}t_{2}}-e^{\frac{\alpha_{2}}{T_{h}}t_{1}})+\ldots ..]$$
where $N_{2}$, $N_{3}$—the calculated power values obtained through multiplying the measured force value by the processing speed, $t_{n}$—the last of the sample of the processing time discrete, $L(t_{n})$—the last value from the sample of the tool path. Here, I note that the adopted approach to describing the relationship between the energy and thermodynamic subsystems of the cutting system fits perfectly into modern ideas about the temperature-force nature of cutting [31].

The results of modeling the discrete version of the integral operator, presented by Equation (10), are given here together with the initial temperature graph measured in the experiment by the STD.201-1 stand. This clearly demonstrates the adequacy of the mathematical apparatus developed and presented in the paper earlier (see Figure 7).

In Figure 7, the simulated temperature characteristic is shown in red, and the temperature characteristic measured in the experiment is shown in black. Here, integral operator (2), presented by discrete Equation (10), adequately reflects the dependence of the temperature in the tool–workpiece contact zone on the change (increase) of the power of irreversible transformations. It can be seen that in the experiment under consideration, temperature stabilization occurs in a time close to 3 s. At the same time, the maximum mismatch between the simulated and experimental characteristics is observed at the beginning of the graph and in the interval from 15 to 17 s where the difference is almost 45 °C.

The simulation results for the case of vibration measurement on the experimental equipment developed by the author of the paper are shown in Figure 8.

Figure 9 shows the results of temperature measurement by a thermocouple built into the tool (see Figure 1c), as well as the results of simulating the temperature characteristics by Equation (9), considering the value of the VA signal taken in the experiment (about 30 mm/s) and the cutting parameters adopted here.

As can be seen from Figure 9, the temperature measurement by an artificial thermocouple (black line on the graph) shows the temperature rise time in about the same time as in the case shown in Figure 7, about 3 s. However, the temperature characteristic in Figure 7 is still some overshoot, and the characteristic measured by an artificial thermocouple monotonically increases to a certain steady-state value. This is due to the principles of measuring the temperature through the thermo-emf taken from a tool and a spindle with a workpiece fixed in it. Concurrently, at the beginning of the measurements, the thermo-emf will show a higher temperature than it actually is. The fact is that the higher the temperature gradient, the higher the thermo-emf; that is, at the initial moment of time, the temperature in the tool–workpiece contact shows a greater value than in the subsequent time after the stabilization of the process and the propagation of the thermal field from the cutting zone deep into the tool and the workpiece. As a result of the thermal energy dissipation in the tool and the workpiece, the temperature gradient decreases, and the temperature value taken by this method also slightly reduces. In this regard, I believe that the experimental characteristic of the temperature increase presented in Figure 9 more accurately reflects the nature of the temperature increase during the initial processing period.

The graph of the simulated temperature (Figure 9) statistically less accurately reflects the changes in the experimentally determined temperature in the cutting zone, so in the interval from 1 to 2 s, the difference between the graphs reaches 160 °C, and in the interval from 2 to 3 s—140 °C. As for modeling the temperature rise in the cutting zone, the technique based on the use of a discrete version of the modified Volterra operator (see Equation (10)) gives a more accurate temperature value than the method based on the implementation of the same operator under the assumption of stationarity of the power values of irreversible transformations (see Equation (9)). This is clearly seen in the initial part of the temperature characteristic shown in Figure 9, where the discrepancy between the measured temperature value and the observed value is large enough. For greater certainty about this discrepancy, let us consider whether Equation (7) is a solution to the Cauchy problem for the differential equation of thermal conductivity. The thermal conductivity equation for this case will take the following form [32,33]:(12)$$\frac{dQ_{}}{dt}=\lambda^{2}\frac{\partial^{2}Q}{\partial L^{2}}+V_{c}\frac{\partial Q}{\partial L}$$
where *Q* (*L, t*)—the function that sets the temperature at a point with coordinate *L* at time *t*. When inserting (7) into the differential equation of thermal conductivity (11), obtain: (13)$$\alpha_{1}V_{c}e^{-\frac{\alpha_{1}}{\lambda }L}(1-e^{-\frac{\alpha_{2}}{T_{h}}t})+\alpha_{2}e^{-\frac{\alpha_{2}}{T_{h}}t}(1-e^{-\frac{\alpha_{1}}{\lambda }L})=-{\left(\frac{\alpha_{1}}{\lambda }\right)}^{2}e^{-\frac{\alpha_{1}}{\lambda }L}(1-e^{-\frac{\alpha_{2}}{T_{h}}t})+\frac{\alpha_{1}}{\lambda }V_{c}e^{-\frac{\alpha_{1}}{\lambda }L}(1-e^{-\frac{\alpha_{2}}{T_{h}}t})$$
or resolving with respect to time and distance:(14)$$\frac{\alpha_{2}e^{-\alpha_{2}t}}{T_{h}(1-e^{-\alpha_{2}t})}=\frac{-\alpha_{1}{}^{2}e^{-\alpha_{1}L}}{\lambda^{2}(1-e^{-\alpha_{1}L})}$$

The analysis of Equation (13) shows that the stationary temperature development option proposed in Equation (7) in the tool–workpiece contact zone is valid only for large values of time (*t*), due to the accepted stationary motion of the temperature source *L = Vt*. This is partly due to the fact that, in the case of metalworking, the approximation of the temperature field to a stationary state is possible only after some transient process associated with the penetration of the tool into the workpiece. Alongside this, the time of establishing a certain quasi-stationary state in case of the measured characteristic and the stationary state in case of the simulated characteristic of the temperature value coincide (see Figure 9).

The measurement and simulation results presented in Figure 9 allow us to determine the tool flank wear rate based on the analysis of the parameters of Equation (9) obtained under modeling. By these parameters, understand the time constant of the thermodynamic system $T=\sqrt{\frac{\lambda h_{3}}{VA\alpha_{1}\alpha_{2}V_{c}}}$ and the gain of this system $k=\frac{k_{Q}\lambda h_{3}}{VA\alpha_{1}\alpha_{2}V_{c}}$. In the presented simulation case (see Figure 9), wear value *h*_3_ was about 0.1 mm. This value was determined experimentally from an enlarged photograph of the trailing edge of the cutting plate (see Figure 2). However, the values of these constants were obtained using scaling coefficients $\alpha_{1}\alpha_{2}$, that were not known in advance. In this regard, in practice, to assess the tool flank wear rate, it is required to conduct preliminary studies. That is, at the beginning of processing, when the wear rate is either zero or known, it is necessary to carry out a preliminary penetration of the tool into the workpiece. Then, based on the results, select the values of these scaling coefficients through comparing experimental and simulated characteristics. After that, these values can be used in the future without changes. In this case, the time constant of the thermodynamic subsystem of the cutting system is conveniently specified using the method of identifying the time constant of the second-order inertial link [34]. The value of the transfer factor of the thermodynamic subsystem can be determined from the averaged steady-state value of the experimental temperature characteristic in the tool–workpiece contact zone. By both of these parameters ($T=\sqrt{\frac{\lambda h_{3}}{VA\alpha_{1}\alpha_{2}V_{c}}}$, $k=\frac{k_{Q}\lambda h_{3}}{VA\alpha_{1}\alpha_{2}V_{c}}$), it is possible to calculate the tool flank wear rate ($h_{3}$). To simplify the calculations, you can use only the transfer factor value. For clarification, you can calculate both values and obtain an estimate of wear through averaging these two results. 

In general, the investigation performed is in line with many of the same modern works in the field of describing the interconnected dynamics of metal cutting processes on metal-cutting machines [35]. However, the presented method of mathematical simulation of the temperature in the treatment zone and the technique based on it for the indirect assessment of the tool wear rate are proposed for the first time.

## 5. Conclusions

The main scientific results are the synthesized mathematical model represented by Equation (9), as well as, based on the use of this model, a method for assessing the degree of tool wear by identifiable parameters. At the same time, this method and the model used in it allow us to assess the degree of tool wear by a set of indirectly observed parameters of the cutting process. Such observable parameters are the time of temperature rise in the contact zone of the tool and the workpiece; the final average measured value of the temperature in the processing zone and the RMS value of the power of the vibration signal are taken either from the tool or from the machine support. Due to this, when monitoring the dynamics of the cutting process in order to assess the wear of the cutting wedge, it is possible to rely on averaged design characteristics in calculations, which should exclude the influence of random fluctuations in the processing process. Proceeding from this, as the main scientific position arising from the analysis of the results obtained, I see the following: the evolution (wear) of the tool when cutting metals on metal-cutting machines is fully reflected in the change in the parameters of the thermodynamic subsystem of the cutting system.

From the point of view of the possibility of the practical application of the obtained scientific results, it follows from the direct use of the method of assessing the degree of deterioration of the instrument proposed in this work. For direct measurement and possible calculation of the averaged vibration characteristics, temperature and reaction time of the thermodynamic subsystem, it will be necessary to develop and introduce into modern production an intelligent cutting tool that would include a built-in temperature sensor and a built-in vibration signal converter. The calculation of the parameters of the thermodynamic subsystem should be carried out on a microcontroller built into the tool holder, from which the obtained value of the degree of tool wear can be directly transmitted to the enterprise network, where the management system of the entire enterprise, based on these data, makes decisions on planned operations on specific metal-cutting machines or metalworking centers. The implementation of the method of assessing the degree of deterioration in this form, as I have described, inherently corresponds to the main direction of the development of modern industrial production within the framework of Industry 4.0.

## Figures and Tables

**Figure 1 materials-14-06492-f001:**
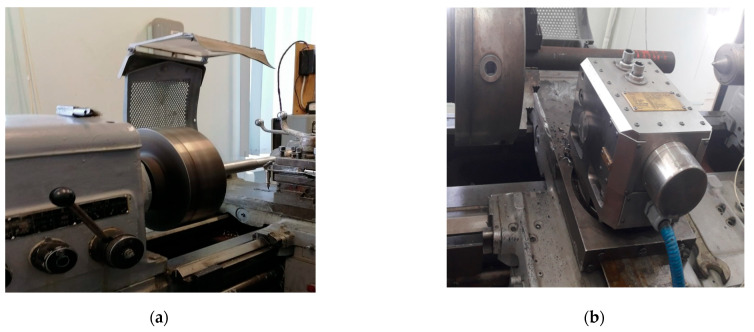
Testing equipment: (**a**) 1k625 machine, (**b**) measuring stand STD. 201, (**c**) measuring complex with a thermocouple built into the tool, (**d**) measuring complex on 1k625 machine, (**e**) measuring system, (**f**) vibration converter AR2081-10.

**Figure 2 materials-14-06492-f002:**
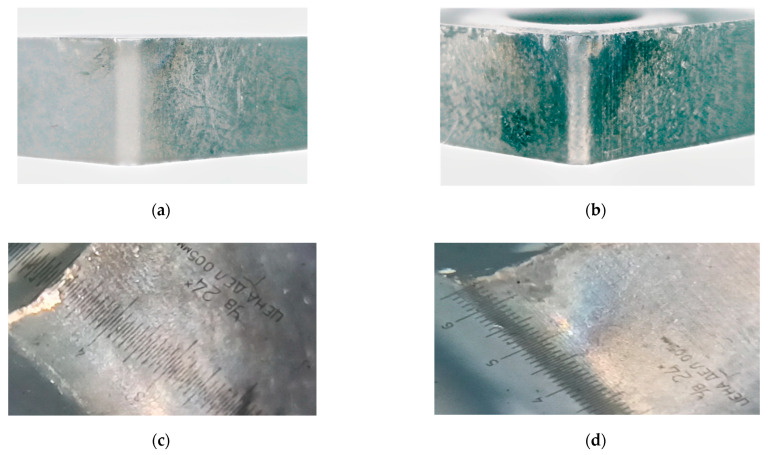
Photos of the cutting plate 10113-110408 T15K6 (**a**,**b**) and WNUM 120,612 (02114-120612) (**c**,**d**).

**Figure 3 materials-14-06492-f003:**
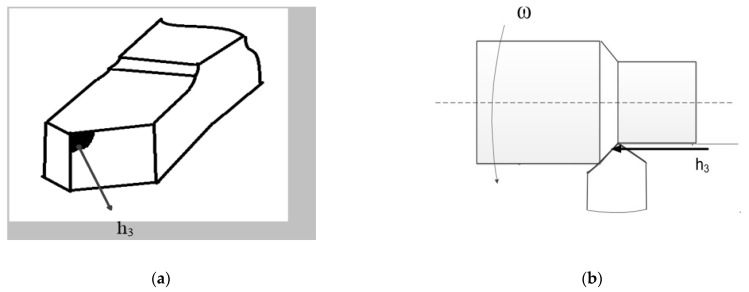
Mechanism of temperature transfer through the back edge of the tool during cutting: (**a**) formed back edge, (**b**) contact of the back edge during cutting.

**Figure 4 materials-14-06492-f004:**
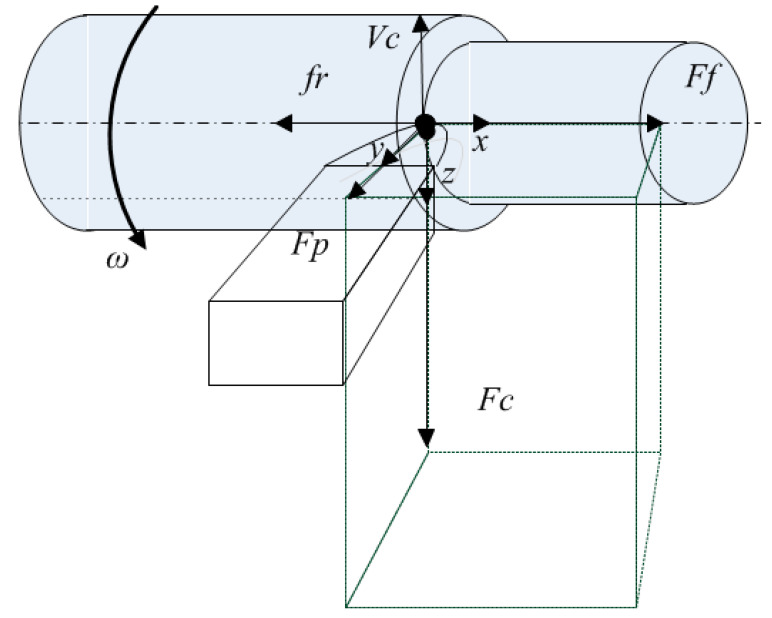
Decomposition of deformations and forces along the axes.

**Figure 5 materials-14-06492-f005:**
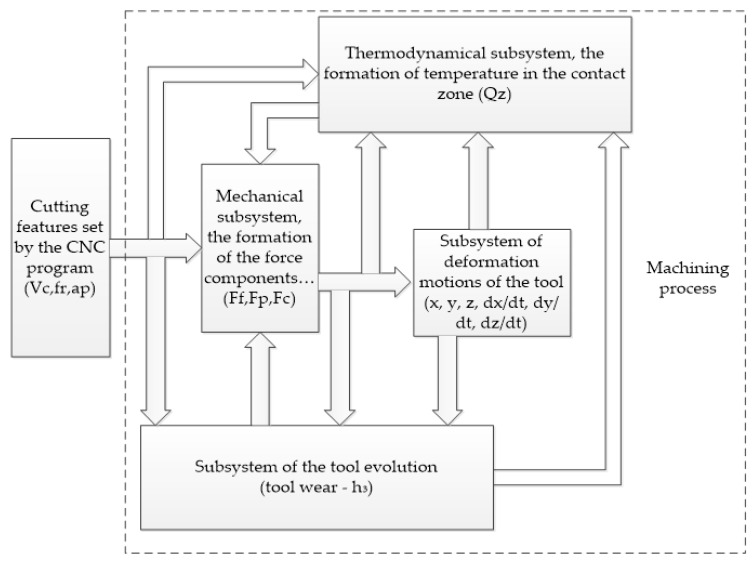
Cutting control system.

**Figure 6 materials-14-06492-f006:**
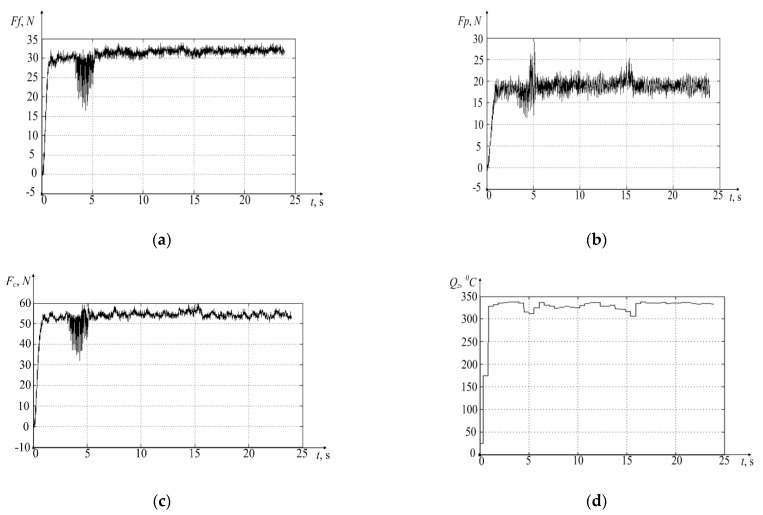
Results of measuring forces along the deformation axes (**a**–**c**) and temperature (**d**).

**Figure 7 materials-14-06492-f007:**
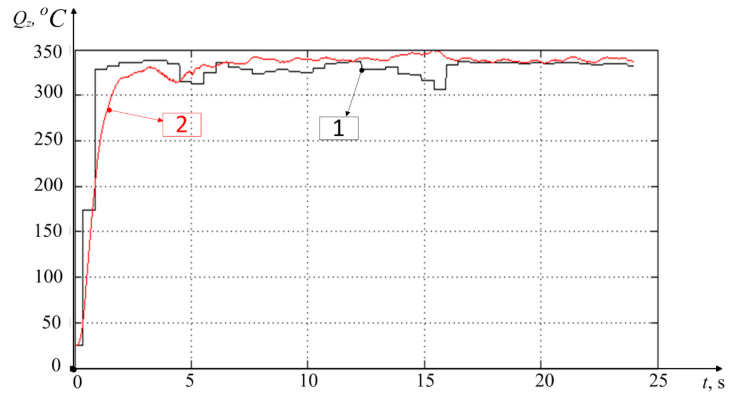
Operator (10) simulation results where 1—experimental characteristic, 2—simulated characteristic.

**Figure 8 materials-14-06492-f008:**
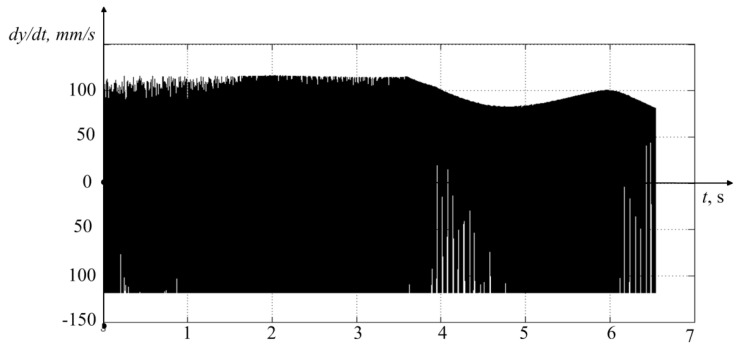
Results of vibration measurement via channely.

**Figure 9 materials-14-06492-f009:**
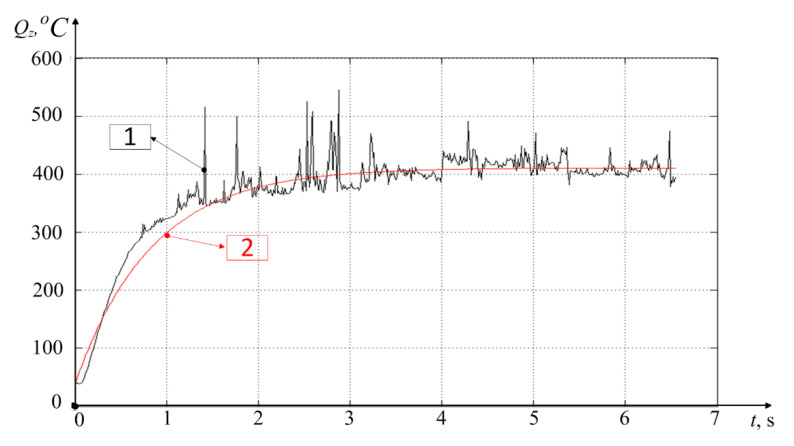
Temperature measurement and simulation results: 1—experimental characteristic, 2—simulated characteristic.

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
