# Peer review of "Turning Tool Wear Estimation Based on the Calculated Parameter Values of the Thermodynamic Subsystem of the Cutting System"

_materials, 2021, doi:10.3390/ma14216492_

Round 1
Reviewer 1 Report
The author's idea is good. However, the article requires major corrections. The suggestions are as follows:
- Technical question. Why do you use labels differently than usual in a complete survey ("V" or "v", "S" or "f", "tp" or "ap", etc.)
- You use the term "we" in many places in the text. Who is "we"? As far as I can see there is only one author of this article. Clarify this.
- Are all figures original or are some of the figures pre-published?
- The abstract is too general written. It should be written much more concisely. The abstract should emphasize: problem, idea, goal, applied methods, scientific benefits, main results, main results.
- The last paragraph in the Introduction section should be corrected. First, write the shortcomings of previous research. Then, based on that, write the goals of your research. Finally, write the scientific benefits and scientific contribution of this research.
- "The mode parameters in the experiments were as follows: cutting speed V=124 m/min, feed S=0.11 mm/rev, cutting depth tр=1 mm for the entire period of the experiment, the cutting parameters remained unchanged." Explain your choice. Elaborate further in the corrected article.
- "The experiment itself and its results are described in more detail in our earlier papers [23-24]." The results, in some form, must be presented and cited. That is missing. It is difficult to follow this research.
- What is a "cutting plate". Is that a good term? What exactly is meant.
- Detailed data on the workpiece (mechanical, physical and thermal characteristics, etc.) should be presented.
- How are equations 1 and 2 obtained? In general, most equations are not clearly presented, not cited, not explained how they were obtained, etc. Clarify everything in detail.
- Further elaborate on the universality of the methodology.
- What fixtures did you use and do they have an impact on the results obtained. Is there an impact of the fixtures (locating and clamping) on the results?
- Data on measuring instrumentation are missing. The parameters of the measurement process should be given. Also show detailed information about the measurement process.
- Analyze and discuss measurement errors.
- The physics or the science behind the experiments needs to be clarified with the interpretation of the results. The results should be scientifically discussed. The results should be compared with the results of previous studies.
- Conclusions should be rewritten. Currently, the results are repeated in the conclusions. The conclusions should highlight the main results, scientific contribution, scientific benefits, possibilities of practical application, limitations of the methodology and future research.
Author Response
Response to the reviewer
I express my deep gratitude to the reviewer for his interest in my work. Below is a description of the results of the processing of the article in accordance with the comments made.
- Technical question. Why do you use labels differently than usual in a complete survey ("V" or "v", "S" or "f", "tp" or "ap", etc.)
Response 1: The relevant comments of the reviewer corrections were made to the work
- You use the term "we" in many places in the text. Who is "we"? As far as I can see there is only one author of this article. Clarify this.
Response 2: The author of the article is really one, but when translated into English, the results given in the third person were described as the results of the work of several authors. The reviewer's comment has been fully taken into account and appropriate substitutions have been made in the text of the article.
- Are all figures original or are some of the figures pre-published?
Response 3: In some of the author's earlier publications, figures of an explanatory nature were presented, in the article these are figures 3,4.
- The abstract is too general written. It should be written much more concisely. The abstract should emphasize: problem, idea, goal, applied methods, scientific benefits, main results, main results.
Response 4: The abstract has been completely rewritten in accordance with the reviewer's comment.
- he last paragraph in the Introduction section should be corrected. First, write the shortcomings of previous research. Then, based on that, write the goals of your research. Finally, write the scientific benefits and scientific contribution of this research/
Response 5: The introduction is supplemented with material in accordance with the reviewer's comment.
- "The mode parameters in the experiments were as follows: cutting speed V=124 m/min, feed S=0.11 mm/rev, cutting depth tр=1 mm for the entire period of the experiment, the cutting parameters remained unchanged." Explain your choice. Elaborate further in the corrected article.
Response 6: The article has been corrected in accordance with the reviewer's comment.
- "The experiment itself and its results are described in more detail in our earlier papers [23-24]." The results, in some form, must be presented and cited. That is missing. It is difficult to follow this research.
Response 7: In the article, the phrase is clarified (changed), and the description of the experiment is significantly expanded.
- What is a "cutting plate". Is that a good term? What exactly is meant.
Response 8: The term "cutting plate" indicated by the reviewer is widely used when describing processes in metalworking. So in the work of Rizzo, A.; Goal, S.; Grille, M.L - 32 number in the list of references, on page 4, Figure 3 is presented, where the authors use this term.
- Detailed data on the workpiece (mechanical, physical and thermal characteristics, etc.) should be presented.
Response 9: The article has been corrected in accordance with the reviewer's comment.
- How are equations 1 and 2 obtained? In general, most equations are not clearly presented, not cited, not explained how they were obtained, etc. Clarify everything in detail.
Response 10: The article has been corrected in accordance with the reviewer's comment, added links to sources.
- What fixtures did you use and do they have an impact on the results obtained. Is there an impact of the fixtures (locating and clamping) on the results.
Response 12: Yes, the article has been additionally amended to reveal the essence of this problem.
- Data on measuring instrumentation are missing. The parameters of the measurement process should be given. Also show detailed information about the measurement process.
Response 13: The article is supplemented with data on measuring systems used in experiments.
- The physics or the science behind the experiments needs to be clarified with the interpretation of the results. The results should be scientifically discussed. The results should be compared with the results of previous studies.
Response 14: The work is supplemented in the part of the introduction and conclusion, where this comment of the reviewer is taken into account.
- Conclusions should be rewritten. Currently, the results are repeated in the conclusions. The conclusions should highlight the main results, scientific contribution, scientific benefits, possibilities of practical application, limitations of the methodology and future research.
Response 14: The conclusions are completely redone in accordance with the remark.
Reviewer 2 Report
The work seems interesting and publishable. However, a few minor adjustments need to be made to the manuscript.
- The introduction lacks a more detailed analysis of what has been done in the area in the past. There is a lack of confrontation of the achieved results of other authors in the given issue.
- Too old literary sources are used in the work.
- The forces Fx, Fy and Fz are non-standardly given in kg in Figure 6.
- Missing list of used symbols and abbreviations.
However, these errors do not significantly reduce the overall quality level of the manuscript. Therefore, it may be published in the journal Materials after minor modifications.
I would like to thank the editor for allowing me to review this type of work and the authors for their research efforts.
Author Response
Response to the reviewer
I express my deep gratitude to the reviewer for his interest in my work. Below is a description of the results of the processing of the article in accordance with the comments made.
- The introduction lacks a more detailed analysis of what has been done in the area in the past. There is a lack of confrontation of the achieved results of other authors in the given issue.
Response 1: The introduction is supplemented with material in accordance with the reviewer's comment.
- Too old literary sources are used in the work.
Response 2: The work contains references to 32 sources, most of which are modern, however, and this is a fair remark, some of the sources were published quite a long time ago. These publications are either monographs [29-30] or review papers [6], the references to which are valid, since basically we are talking about the mathematical representation of temperature effects, which has not changed since then.
- The forces Fx, Fy and Fz are non-standardly given in kg in Figure 6.
Response 3: The article has been corrected in accordance with the reviewer's comment.
- Missing list of used symbols and abbreviations.
Response 4: A list of abbreviations used has been added to the article.

Reviewer 3 Report
Dear Author:
after recognising your manuscript, I would like say that it is very interesting work which is focused on novel method for tool wear estimation. This work is part of the current development trend in industry towards intelligent monitoring of production processes.
However I have a few remarks according this work, which in my opinion should be corrected.
1) The abstract should refer more to the analyses carried out and the results obtained. It should also present the main conclusions. Giving in the abstract an abbreviation describing the measuring station without explanation is incorrect.
2) The work focuses on metalworking processes, but manufacturing and cutting processes are not only concerned with metal materials. In my opinion, a broader presentation of currently used methods for monitoring and predicting tool wear is missing. This should also apply to materials other than metals. Such work as presented below will, in my opinion, be helpful in broadening the analysis of the problem:
Nasir, V., Sassani, F. A review on deep learning in machining and tool monitoring: methods, opportunities, and challenges. Int J Adv Manuf Technol 115, 2683–2709 (2021). https://doi.org/10.1007/s00170-021-07325-7
KuntoÄŸlu, M.; Aslan, A.; Pimenov, D.Y.; Usca, Ü.A.; Salur, E.; Gupta, M.K.; Mikolajczyk, T.; Giasin, K.; KapÅ‚onek, W.; Sharma, S. A Review of Indirect Tool Condition Monitoring Systems and Decision-Making Methods in Turning: Critical Analysis and Trends. Sensors 2021, 21, 108. https://doi.org/10.3390/s21010108
The dynamics of the cutting process is also highly dependent on the appropriate machine tool and spindle design, as demonstrated by Orlowski et al 2020.
Orlowski, K.A.; Dudek, P.; Chuchala, D.; Blacharski, W.; Przybylinski, T. The Design Development of the Sliding Table Saw Towards Improving Its Dynamic Properties. Appl. Sci. 2020, 10, 7386. https://doi.org/10.3390/app10207386
2) Materials and Methods. The author reports that a series of experiments have been carried out. However, there is no information on their number and type. Giving the models of the machine tool does not inform the reader much, there should also be information about the manufacturer and the country of origin. A similar remark concerns the measuring stand.
3) Materials and Methods. Symbols for cutting parameters are not in accordance with the iso standard. These should be as below:
cutting speed - vc
feed per revolution - fr
depth of cut - ap.
4) Materials and Methods. Description of toolholder and insert symbols is not sufficient, the manufacturer and country of origin must be stated. In addition, the geometry of the cutting blades used in the tests must be described in detail.
5) Materials and Methods. The chemical composition of the material tested and the basic mechanical properties must be given, and the standard to which the material symbol is presented must be stated.
An accurate depiction of the position of the accelerometer sensors is missing. Figure 2 should be expanded.
6) The article should include a list of symbols - nomenclature. This will improve the readability of the article.
7) Figure 4. The components of cutting forces should be used to analyse the process, together with a demonstration of their effect on blade wear. Therefore, I propose to change the symbols Fx, Fy, Fz into the specific force components, i.e. feed force Ff, back force Fp, cutting force Fc.
8) line 162 and line 183, same symbol but different parameter. this must be corrected throughout the work for all symbols.
9) Figure 6. Forces have wrong units, they should be expressed in N
10) statistical significance analysis of the differences between the values obtained experimentally and using the model is missing
11) The conclusions are too general and do not relate precisely to the results obtained.
Author Response
Response to the reviewer
I express my deep gratitude to the reviewer for his interest in my work. Below is a description of the results of the processing of the article in accordance with the comments made.
- The abstract should refer more to the analyses carried out and the results obtained. It should also present the main conclusions. Giving in the abstract an abbreviation describing the measuring station without explanation is incorrect.
Response 1: The abstract has been corrected in accordance with the comment made by the reviewer.
- The work focuses on metalworking processes, but manufacturing and cutting processes are not only concerned with metal materials. In my opinion, a broader presentation of currently used methods for monitoring and predicting tool wear is missing. This should also apply to materials other than metals. Such work as presented below will, in my opinion, be helpful in broadening the analysis of the problem.
Response 2: The work, in the Introduction, has been supplemented in accordance with the reviewer's comment, and an additional source proposed by the reviewer has also been added to the list of references [7].
- Materials and Methods. The author reports that a series of experiments have been carried out. However, there is no information on their number and type. Giving the models of the machine tool does not inform the reader much, there should also be information about the manufacturer and the country of origin. A similar remark concerns the measuring stand.
Response 3: Appropriate additions have been made to the work, revealing information about the experiment and equipment..
- Materials and Methods. Symbols for cutting parameters are not in accordance with the iso standard. These should be as below.
Response 4: The paper replaces the symbols used in accordance with the reviewer's comment.
- Materials and Methods. Description of toolholder and insert symbols is not sufficient, the manufacturer and country of origin must be stated. In addition, the geometry of the cutting blades used in the tests must be described in detail.
Response 5: The article is supplemented with materials revealing information about the tool.
- Materials and Methods. The chemical composition of the material tested and the basic mechanical properties must be given, and the standard to which the material symbol is presented must be stated.
Response 6: The characteristics of the alloy are disclosed in the corrected version of the article, and analogs of the alloy according to American and German standards are also given.
- An accurate depiction of the position of the accelerometer sensors is missing. Figure 2 should be expanded.
Response 7: Figure 1 has been supplemented.
- The article should include a list of symbols - nomenclature. This will improve the readability of the article.
Response 8: The article is supplemented with a list of symbols in accordance with the remark.
- Figure 4. The components of cutting forces should be used to analyse the process, together with a demonstration of their effect on blade wear. Therefore, I propose to change the symbols Fx, Fy, Fz into the specific force components, i.e. feed force Ff, back force Fp, cutting force Fc.
Response 9: The drawing has been corrected in accordance with the reviewer's comment.
- line 162 and line 183, same symbol but different parameter. this must be corrected throughout the work for all symbols.
Response 10: Changes have been made to the article.
- Figure 6. Forces have wrong units, they should be expressed in N.
Response 11: Figure 6 has been corrected in accordance with the remark.
- Statistical significance analysis of the differences between the values obtained experimentally and using the model is missing.
Response 12: There is no statistical analysis due to the fact that I propose to calculate the degree of tool wear by averaged measured characteristics such as the reaction time of the thermodynamic subsystem to changes in the power of irreversible transformations and the average value of the measured temperature signal. It is for this reason that no statistical analysis of experimental data is carried out in the work, the results of this analysis are not used in the proposed method.
- The conclusions are too general and do not relate precisely to the results obtained.
Response 13: The conclusions of the work have been completely redone in accordance with the reviewer's comment.

Round 2
Reviewer 1 Report
The article has been corrected.
Author Response
Thank you for your work in evaluating my article
Reviewer 3 Report
Dear Author,
thank you for your responses to my comments. I regret to say that although changes have been made they are not sufficient and still do not bring the article to a publishable level. In addition, the author in his answers does not indicate exactly how he made the improvement and how this part of the work currently presents itself. Repeatedly the Author states that changes have been made, when in fact they have not. Such a state is unacceptable in the editorial-review process in a reputable journal. Such behaviour is unethical and should be decisively eliminated by the publisher. Please find below my main comments
1) Abstract is still rather laconic and does not clearly refer to the results achieved and conclusions.
2) Introduction is still an underdeveloped part of work. The processes of predicting the wear of cutting tools have recently been very widely analysed by the scientific community. Also the analyses of the influence of dynamic parameters of the cutting process and system stiffness have been analysed in many papers. These issues should, in a way, constitute an introduction to the issue analysed by the Author, and here we do not see such an introduction supported by appropriate references.
3) Line 237. the angle of attack symbol should have a lower orthogonal plane index "o". What does the parameter φ mean? Both the text and the nomenclature list lack an explanation of this parameter? The same is true of the symbol λ, which in the list of nomenclature is described as a coefficient, while in line 237 it denotes a geometrical parameter of the cutting edge. This should be corrected throughout the work.
4) All standards cited should be included in the list of references
5) There is no information on the finish of the semi-finished (shaft). Whether it was made using hot rolling or drawing technology, for example. This affects the machinability properties and such information should be included in the scientific work, not only the chemical composition of steel.
6) The author added a list of symbols-nomenclatures, but unfortunately it is very poor and does not represent all the symbols used in the work. In addition, there are duplicate symbols in the work, which makes the work very difficult to read for the reader. This should be corrected by a very thorough and meticulous analysis of the entire work.
7) The author claims in his reply that he has taken into account my comment #10 from the first review, which is not true, as these symbols are still duplicated.
8) If the models have been created based on physical measurements (vibration, acceleration, temperature) then I understand that their respective levels have been related to the actual wear of the cutting blade. Models created in this way should then be verified in a further experimental trial. The results of this trial should be analysed statistically for significance of differences between the results of the model and the results of the verification experiment. This would be the correct scheme.
9) The proposals are still too general. They do not summarise the results obtained. Does the model allow statistically significant prediction of cutting blade wear? What errors does the proposed model have with respect to actual wear? Are the physical parameters measured in the experiment sufficient to monitor the condition of the cutting edge, or does this method have any limitations, etc.?
Author Response
Response to the reviewer
Once again, I express my deep gratitude for the interest in my work. Below are the answers to the reviewer's questions.
- Abstract is still rather laconic and does not clearly refer to the results achieved and conclusions.
Response 1: Unfortunately, the requirements of the reviewers in this matter contradict each other. The first reviewer demands conciseness in writing the abstract, the third indicates that the volume of the abstract is insufficient. The requirements of the journal limit the volume of the abstract, no more than 200 words. Therefore, I was able to add only one sentence revealing the results to the abstract of the article.
- Introduction is still an underdeveloped part of work. The processes of predicting the wear of cutting tools have recently been very widely analysed by the scientific community. Also the analyses of the influence of dynamic parameters of the cutting process and system stiffness have been analysed in many papers. These issues should, in a way, constitute an introduction to the issue analysed by the Author, and here we do not see such an introduction supported by appropriate references.
Response 2: In part of the introduction, I have analyzed the sources describing the problems associated with this remark. In the introduction, material has been added that reveals these issues more deeply.
- Line 237. the angle of attack symbol should have a lower orthogonal plane index "o". What does the parameter φ mean? Both the text and the nomenclature list lack an explanation of this parameter? The same is true of the symbol λ, which in the list of nomenclature is described as a coefficient, while in line 237 it denotes a geometrical parameter of the cutting edge. This should be corrected throughout the work.
Response 3: Indeed, I made a mistake here, which I did not pay attention to earlier, to be honest, I simply did not understand the previous version of the question. I apologize for this point. The article contains corrections corresponding to this remark.
- All standards cited should be included in the list of references.
Response 4: All standards are included with the list of references.
- There is no information on the finish of the semi-finished (shaft). Whether it was made using hot rolling or drawing technology, for example. This affects the machinability properties and such information should be included in the scientific work, not only the chemical composition of steel
Response 5: The shaft made of 45 steel was made using hot rolling technology, before the experiment the shaft was cut to a length of 75 centimeters, then there was a pre-finishing treatment of the shaft surface and precise alignment. This information is included in the text of the article.
- The author added a list of symbols-nomenclatures, but unfortunately it is very poor and does not represent all the symbols used in the work. In addition, there are duplicate symbols in the work, which makes the work very difficult to read for the reader. This should be corrected by a very thorough and meticulous analysis of the entire work
Response 6: the list of characters used has been expanded, duplicate characters have been removed from the text of the article.
- The author claims in his reply that he has taken into account my comment #10 from the first review, which is not true, as these symbols are still duplicated.
Response 7: I apologize once again for this point of comments, in the previous version of the review I misinterpreted this question. Appropriate corrections have now been made to the article.
- If the models have been created based on physical measurements (vibration, acceleration, temperature) then I understand that their respective levels have been related to the actual wear of the cutting blade. Models created in this way should then be verified in a further experimental trial. The results of this trial should be analysed statistically for significance of differences between the results of the model and the results of the verification experiment. This would be the correct scheme
Response 8: Unfortunately, today I cannot give a sufficient statistical assessment of the obtained models, experiments on this equipment are still ongoing and new options are emerging that require consideration in the model parameters. The static evaluation will be carried out later and described in the following articles..
- The proposals are still too general. They do not summarise the results obtained. Does the model allow statistically significant prediction of cutting blade wear? What errors does the proposed model have with respect to actual wear? Are the physical parameters measured in the experiment sufficient to monitor the condition of the cutting edge, or does this method have any limitations, etc.?
Response 9: To date, work is underway on the creation of an industrial design of the tool complex with its subsequent implementation in the operating machine-building organization. Based on the results of the implementation, it is planned to obtain a sufficient amount of data to assess the statistical accuracy of the model and method, as well as to fully determine the limitations affecting it. Unfortunately, laboratory tests are not enough for this.
